# Ascorbate Is a Primary Antioxidant in Mammals

**DOI:** 10.3390/molecules27196187

**Published:** 2022-09-21

**Authors:** Junichi Fujii, Tsukasa Osaki, Tomoki Bo

**Affiliations:** 1Department of Biochemistry and Molecular Biology, Graduate School of Medical Science, Yamagata University, Yamagata 990-9585, Japan; 2Laboratory Animal Center, Institute for Promotion of Medical Science Research, Yamagata University Faculty of Medicine, Yamagata 990-9585, Japan

**Keywords:** ascorbate, GULO, RGN, AKR1A, antioxidant, radical scavenging

## Abstract

Ascorbate (vitamin C in primates) functions as a cofactor for a number of enzymatic reactions represented by prolyl hydroxylases and as an antioxidant due to its ability to donate electrons, which is mostly accomplished through non-enzymatic reaction in mammals. Ascorbate directly reacts with radical species and is converted to ascorbyl radical followed by dehydroascorbate. Ambiguities in physiological relevance of ascorbate observed during in vivo situations could be attributed in part to presence of other redox systems and the pro-oxidant properties of ascorbate. Most mammals are able to synthesize ascorbate from glucose, which is also considered to be an obstacle to verify its action. In addition to animals with natural deficiency in the ascorbate synthesis, such as guinea pigs and ODS rats, three strains of mice with genetic removal of the responsive genes (GULO, RGN, or AKR1A) for the ascorbate synthesis have been established and are being used to investigate the physiological roles of ascorbate. Studies using these mice, along with ascorbate transporter (SVCT)-deficient mice, largely support its ability in protection against oxidative insults. While combined actions of ascorbate in regulating epigenetics and antioxidation appear to effectively prevent cancer development, pharmacological doses of ascorbate and dehydroascorbate may exert tumoricidal activity through redox-dependent mechanisms.

## 1. Introduction

Ascorbate (Asc) was first discovered in the adrenal gland [1], but, even after one century, our knowledge of the many diverse functions of Asc continues to increase. While Asc acts as an electron donor in the synthesis of biological compounds, such as collagen and catecholamines [2], it also functions as a cofactor for the oxygen-dependent regulation of gene expression via the hypoxia-inducible factor (HIF)-1α [3] and for the epigenetic regulation of genes concerning the methylation status of both DNA and histones [4,5].

In addition to its roles in enzymatic reactions, antioxidation that mainly proceeds in a non-enzymatic manner in animals is another pivotal function of Asc [6]. The daily ingestion of massive amounts of Asc, compared to other micronutrients, are required for maintaining health in Asc-incompetent animals. In addition to animals with a natural Asc-deficiency, such as the guinea pig, studies on Asc-incompetent mice by means of genetic modification have revealed that the antioxidative potential of Asc indeed plays important roles in maintaining animal health [7]. On the other hand, due to the high reduction–oxidation (redox) potential, Asc may also exert pro-oxidant effects in the presence of transition metal ions; iron, in most cases [8]. These antithetical properties of Asc have attracted considerable attention in regard to the prevention of and therapeutic applications for cancer [9].

In this review article, we attempt to overview roles of Asc as an antioxidant and also as a pro-oxidant under in vivo conditions, with an emphasis on the pathophysiology of genetically modified, Asc-incompetent mice and on cancer biology, so readers are referred to recent review articles [3,4,5,10] in order to understand the functional aspects of Asc in supporting enzymatic reactions, including hypoxic responses and epigenetics.

## 2. Roles of Asc in Cellular Homeostasis in Mammals

### 2.1. Molecular Dynamics of Asc in the Body

The chemical nature of Asc has been extensively overviewed elsewhere [7,11], and, hence, the structure and chemical properties of Asc, which are essential for understanding its physiological functions, are reviewed only minimally. The fully reduced monoanion (ascorbate; Asc) is the dominant form in the animal body compared to the ascorbyl radical (monodehydroascorbate, Asc•) and the fully oxidized form (dehydroascorbate; DHA) (Figure 1). When Asc participates in enzymatic reactions, two electrons are usually abstracted, resulting in the formation of DHA. In the course of antioxidation, however, Asc preferentially reacts with radical species and is converted into Asc•, which is a radical form but, due to the resonance stabilization of this species, it is less reactive to other compounds compared with oxygen radicals [12]. The in vitro reaction between Asc and hydrogen peroxide (H_2_O_2_) is popularly described but, in fact, it is not very efficient (rate constant k = 2 M^−1^s^−1^) [13]. Asc is much more reactive to a radical species, which allows Asc to act as a potent antioxidant.

Nutritionally, both Asc and DHA are taken up by intestinal cells and utilized as micronutrients [14]. The half-life of Asc is approximately one day in the human body, mainly due to the high water solubility of the molecule. Asc is largely stored within cells, with the plasma/extracellular fluid and inside cells being 0.04–0.08 mM and 0.3–10 mM, respectively, in human [15]. The Na^+^-dependent vitamin C transporter (SVCT) family is responsible for the uptake of Asc by cells and this explains the remarkable difference between extracellular and intracellular Asc concentrations [16,17]. SVCT1 (also referred to as solute carrier family 23 member 1; Slc23a1) is present in the intestine, kidney and liver cells and is involved in both the absorption of Asc from nutritional sources in the intestine and the reabsorption from filtrates in the kidney. Tumor necrosis factor-α (TNF-α) suppresses the transcription of the SVCT1 gene via the nuclear factor-κB (NF-κB) pathway, which results in the inhibition of the intestinal uptake of Asc [18]. SVCT2 (Slc23a2), which is expressed in the brain, accounts for the uptake of Asc from the cerebrospinal fluid [19]. Since it has been suggested that Asc has pivotal neuronal functions, SVCT2 may become a therapeutic target for the treatment of neurodegenerative diseases, such as Alzheimer’s disease, Parkinson’s disease and Huntington’s disease [20]. On the other hand, facilitative glucose (Glc) transporters are involved in the uptake of DHA by cells or cellular organelles. Asc• and DHA are either reductively recycled to Asc via enzymatic (ascorbic acid reductase; AAR) and non-enzymatic routes, metabolized by several processes in cells, or excreted from cells [2].

### 2.2. Asc-Involved Enzymatic Reactions

The reducing power of Asc is utilized for some enzymatic reactions that are involved in the synthesis of compounds, such as monoamines and steroid hormones, and the post-translational modification of proline residues in collagen [2]. Support of Asc on enzymatic reactions extends to controlling hypoxia-induced gene expression through the regulation of HIF-1α levels [3] and the epigenetic regulation of genes [4]. Non-heme iron, either free or in enzyme-bound form, functions cooperatively with Asc in many of these reactions. A group of enzymes, namely α-ketoglutarate-dependent non-heme iron dioxygenases, commonly utilize Asc in their enzymatic reactions. This enzyme group includes prolyl 4-hydroxylase, which is responsible for collagen synthesis and prolyl 4-hydroxylase domain-containing enzymes (PHDs) that regulate HIF-1α function [3,21]. The animal body contains a huge amount of collagen molecules, which constitute ~30% of the total protein mass in the body. Proline residues of collagen molecules are highly hydroxylated (~100 hydroxyprolines in each molecule) and are required for forming the proper structure of triple-stranded helices and secretion from cells [22]. Three types of prolyl 4-hydrolases in the endoplasmic reticulum (ER) are involved in proline hydroxylation during collagen synthesis. They consist of two subunits; the α subunit, which is unique to each subtype and has proline hydroxylase activity, and the β subunit, which is identical to protein disulfide isomerase (PDI) and is common among these subtypes [21]. In addition to Asc, iron, molecular oxygen and α-ketoglutarate are frequently used as co-substrates/co-factors for proline hydroxylation. As the result of an enzymatic activity, DHA, carbon dioxide and succinate are produced. Asc insufficiency results in aberrant collagen synthesis, which may lead to the development of scurvy [23]. Although it has been suggested that Asc has a role in oxidative protein folding in the ER, the donation of electrons to proline hydroxylation in collagen synthesis appears to be the bona fide function of Asc in the ER rather than oxidative protein folding, where DHA acts as the electron acceptor.

HIF-1α is a transcriptional factor and the master regulator of genes whose expression consequently increases oxygen availability [24]. Under normoxic conditions, the PHD-supported hydroxylation of specific prolyl residues in HIF-1α leads to the generation of a binding site for the von Hippel-Lidau (pVHL) syndrome protein. Asc also provides reducing equivalents in the formation of hydroxyproline. pVHL then polyubiquitinates HIF-1α by acting as a ubiquitin ligase and consequently leads to the degradation of HIF-1α by proteasomes. Under hypoxic conditions, a decline in oxygen concentration decreases prolyl hydroxylase activity, which results in the stabilization of HIF-1α, thus allowing its translocation to the nucleus. While the development of a solid cancer depends on the supply of oxygen and nutritional status, pVHL functions to maintain HIF-1α at low levels and suppress neovascularization, thereby acting as a tumor suppressor [25]. This mechanism may partly be responsible for the Asc-mediated prevention of tumor development. Extensive oxidation of prolyl hydroxylase 2 causes dimerization and inactivation, which results in the stabilization of HIF-1α even under normoxic conditions [26]. Thus, prolonged oxidative stress, such as inflammatory conditions, may consume Asc and increase the risk of tumor development through sustained HIF-1α activation.

### 2.3. Asc in the Epigenetic Regulation of Gene Expression

Advances in research on the epigenetic regulation of genes has revealed some novel actions of Asc. Ferrous iron (Fe^2+^)- and α-ketoglutarate-dependent dioxygenases that catalyze the demethylation of DNA and histones also require Asc as an electron-donating cofactor [27]. DNA methyltransferases catalyze methylation at the 5-position of cytosine (5mC) by means of S-adenosyl methionine (SAM), a methyl group donor that is produced during the course of methionine metabolism. At the promoter region of transcriptionally inactive genes, 5mC is dominantly present . This issue is particularly important in cancer biology because the transcriptional suppression of tumor suppressor genes by hypermethylation may increase the risk for carcinogenesis [28].

Furthermore, 5mC is converted into 5-hydroxymethylcytosine by catalysis of methylcytosine dioxygenases ten-eleven translocation (TET) with the help of Asc [29]. During the oxygenase reaction, the Fe^2+^ in TET is oxidized to ferric iron (Fe^3+^), which is then reduced back to Fe^2+^ by Asc. The resulting 5-hydroxymethylcytosine is then converted into 5-carboxylcytosine and is subsequently replaced by an unmodified cytosine through a base excision DNA repair system [30]. Accordingly, Asc-supported TET reactions can reactivate genes that are epigenetically silenced due to the methylation of cytosine [4]. This epigenetic action of Asc may also be responsible for maintaining the pluripotency of stem cells. Multiple genes are transcriptionally active in embryonic stem (ES) cells, and, in fact, the demethylation of DNA by TET can occur in ES cells [31]. For another example, Asc enhances the efficiency of the generation of induced pluripotent stem cells (iPSCs) from both human and mouse origins [32].

Asc is also involved in the demethylation of proteins, notably histones. The Jumonji C-domain-containing histone deaminase 1 (JHDM1) catalyzes the specific demethylation of histone H3 at lysine 36 (H3K36) [33]. The Asc-dependent H3K36 demethylase activity of JHDM1a/1b, in conjunction with the oxidation of three Asc molecules to DHA, results in the regeneration of lysine from trimethyl lysine and leads to the reprogramming of mouse embryonic fibroblasts into iPSCs through the re-activation of genes [34]. Currently, there are approximately 20 proteins in this family that functions to catalyze the removal of methyl groups from methyl lysine with the aid of Asc [35]. Thus, Asc plays essential roles in activating gene expression through donating electrons to the demethylation of both DNA and histones. It should be noted, however, that instead of Asc, a variety of other molecules have also been reported to act as electron donors in these reactions [36], which may compensate for an Asc insufficiency.

## 3. Antioxidant Action of Asc

Oxidative stress that is caused by elevated reactive oxygen species (ROS) with a relatively low antioxidant capacity is associated with unhealthy conditions and aging. While in vitro studies have demonstrated reproducible and reliable antioxidative actions of Asc, ambiguity remains regarding the antioxidant potential of Asc due to the presence of multiple oxidants and antioxidants in vivo. As precise chemical reactions are overviewed repeatedly [6,7], we outline properties of Asc that explain the biological action of Asc here.

### 3.1. Radical-Scavenging Action of Asc

While plants produce Asc-dependent peroxidases, higher animals do not. Thus, antioxidant effects of Asc in animals solely depend on non-enzymatic reactions with ROS. While a variety of nutritional and intrinsically synthesized compounds are involved in antioxidation in the sense of non-enzymatic processes, Asc together with tocopherol (Toc, also called vitamin E) are micronutrients with antioxidative ability. ROS cause oxidative stress and damage to susceptible molecules, such as proteins with reactive cysteine sulfhydryl (Cys-SH), bases in nucleotides, and polyunsaturated fatty acids. Superoxide (O_2_•^−^) is the oxygen radical that is produced first during the metabolic conversion of oxygen molecules in most cases [37]. O_2_•^−^ reacts with limited numbers of compounds and is generally regarded as less potent than hydroxyl radicals (HO•) or other ROS in the body. However, unpaired electrons may be transferred to other molecules and act as initiators for radical chain reactions, which consequently causes the production of more toxic radical species, such as hydroxyl radicals. Thus, the elimination of O_2_•^−^ by superoxide dismutase (SOD) or other antioxidants is considered to be of primary importance for living organisms.

Despite the potential importance of SOD in the elimination of the radical electron of O_2_•^−^, SOD1^−/−^ mice show subtle phenotypic defects [38]. Embryonic fibroblasts isolated from the SOD1^−/−^ mice, however, cannot be maintained under normal oxygen conditions [39]. In another example, SOD1^−/−^ mouse zygotes do not develop beyond the two-cell stage but can be rescued by supplementation with a physiological concentration of Asc [40]. Based on these observations, we conclude that the presence of Asc may be one of the factors that obscure the toxicity of O_2_•^−^ in the SOD1^−/−^ mice. Consistently, a decrease in Asc levels is observed in the mice lacking SOD1 and SOD3, the latter of which is dominantly localized at the lung [41]. The extent of the decrease in Asc is the most severe in a double deficiency, i.e., deficiencies in both SOD1 and SOD3.

Upon interaction with O_2_•^−^, Asc• is further oxidized to DHA by the action of another radical species (Figure 2A), and the resulting H_2_O_2_ can be converted into water by peroxidases, such as glutathione peroxidase (GPX), peroxiredoxin (PRDX), and catalase. The second-order rate constant (k) for the reaction between Asc and O_2_•^−^ is 1 × 10^5^ M^−1^s^−1^ [42]. Although the O_2_•^−^-scavenging potential of Asc is much less than the SOD-catalyzed dismutation (k = 2 × 10^9^ M^−1^s^−1^) [37], the presence of abundant levels of Asc inside cells would allow Asc to act as an important O_2_•^−^ scavenger. The second-order rate constant for Asc• and O_2_•^−^ is 2.6 × 10^8^ M^−1^s^−1^, indicating a much higher reactivity for Asc• than Asc [7,43]. Asc also reacts with other radical species, including the hydroxyl radical HO• (~× 10^10^ M^−1^s^−1^) (Figure 2B), the tocopheroxyl radical (~10^7^ M^−1^s^−1^) (Figure 2C), the alkoxyl radical LO• (~2.0 × 10^6^ M^−1^s^−1^), and lipid peroxyl radicals LOO• (~10^9^ M^−1^s^−1^) (Figure 2D) [6,12]. Toc is a primary scavenger of LO• and LOO• and becomes Toc• (10^3^~7 × 10^6^) [44]. Asc is a highly hydrophilic compound but is able to reduce hydrophobic Toc• back by donating an electron, although the rate constant varies depending on the experimental conditions used (k = 10^6^ ~ 10^7^ M^−1^s^−1^) [8]. Accordingly, Asc and Toc are major nutritional antioxidants that, when acting together, coordinately prevent lipid peroxidation by terminating the radical chain reaction and subsequent oxidative damage [11].

### 3.2. Asc-Involved Reductive Recycling of Reactive Sulfhydryl Groups in Proteins

Hyper reactive Cys-SH that is frequently found in the catalytic center of enzymes tends to form a cysteine thiolate anion (Cys-S^−^), depending on the microenvironment in the protein structure, and is susceptible to oxidation to cysteine sulfenic acid (Cys-SOH) by H_2_O_2_ [45]. Due to oxidative modification of phosphotyrosine phosphatases, phosphorylation-mediated signaling reactions for growth factor receptors are prolonged by H_2_O_2_ via a redox-dependent mechanism [46]. For the termination of the signaling, Cys-SOH is reduced back to Cys-S^−^, although the reductants that are responsible for this are not clearly understood at this time. Glutathione (GSH) is a likely candidate for serving as a reductant that acts on Cys-SOH units in proteins and undergoes glutathionylation [47] (Figure 3). Glutaredoxin (GRX) reductively removes GSH moieties from glutathionylated proteins by employing another GSH, which regenerates Cys-S^−^ [48]. This overall reaction results in the conversion of H_2_O_2_ to water and the oxidation of two GSH molecules to a GSH dimer linked with a disulfide bond (GSSG), as observed in glutathione peroxidase reactions. Asc can directly reduce Cys-SOH back to Cys-SH in some proteins, including glyceraldehyde 3-phosphate dehydrogenase and cysteine proteases, such as papain [49]. According to the rate constant (k = ~10^3^ M^−1^s^−1^), this reaction is not very efficient, but the presence of high concentrations of Asc in cells may make this reaction more feasible.

PRDX possesses a reactive Cys-SH at the catalytic center and SOH is transiently formed during the peroxidase reaction, while thioredoxin instead of GSH is the electron donor for most members of the family of proteins [50]. However, PRDX6 is one Cys-containing PRDX family of proteins that uses GSH as an electron donor. It is also noteworthy that Asc can act as a reductant for Cys-SOH, which is produced as a reaction intermediate in PRDX6 [51]. Thus, some proteins appear to have Asc-dependent peroxidase properties in animals, which may be associated with antioxidative actions of Asc along with direct radical scavenging.

## 4. Pro-Oxidant Properties of Asc

The antioxidant potential of Asc is attributed to its potent electron-donating ability, but electron donation to metal ions, notably iron ions, may stimulate further oxidation reactions [8]. In fact, Asc functions both as an antioxidant and as a pro-oxidant depending on the microenvironment, which results in antithetical actions. However, the pro-oxidant action of Asc is not always harmful but may exert beneficial functions, as evidenced by a pharmacological dose of Asc that is used in anti-cancer chemotherapy [52].

### 4.1. Asc in Iron Homeostasis

Enzymes responsible for Asc-involved reactions largely utilize iron in electron transfer reactions during catalysis. In fact, the physiological actions of Asc are closely associated with iron homeostasis. Asc mediates the reduction of Fe^3+^ to Fe^2+^, which is then taken up by the divalent metal ion transporter DMT1 in apical membranes of the intestinal lumen [53]. The presence of insufficient Asc in the intestinal lumen may be associated with a decrease in iron absorption, resulting in an iron-deficient condition, as represented by anemia [54]. Asc stimulates the accumulation of iron within cells by increasing ferritin levels and inhibits the efflux of iron from cells through suppressing the lysosomal degradation of iron-bound ferritin [55]. Thus, Asc and iron constitute an interrelated system in vivo.

### 4.2. Generation of ROS Via the Redox Cycling of Asc

There has been extensive discussion of the issue of whether Asc exerts antioxidative action with respect to oxidative DNA modification in humans. An increase in oxidized DNA has been reported in healthy volunteers who have taken 500 mg of Asc/day for 6 days [56]. In the presence of H_2_O_2_ in vitro, free iron or copper is responsible for the damage of bases in DNA, and the addition of Asc further enhances the extent of oxidative DNA damage [57]. When healthy subjects receive co-supplements of iron and Asc, the levels of oxidative DNA damage are elevated after 6 weeks [58]. The interaction of Asc with free iron causes a potentially harmful situation in that ROS production is stimulated [55]. Iron is a dominant microelement but is mostly present in either heme-bound form or stored in the ferritin-bound Fe^3+^ form. Fe^3+^ is not toxic per se, but Fe^2+^, which is produced by accepting an electron from a reductant or a radical species, exerts strong toxicity. Asc reduces Fe^3+^ to Fe^2+^, which then converts molecular oxygen and H_2_O_2_ to O_2_•^−^ and HO•, respectively, by donating an electron (Figure 4). The resulting Asc• or DHA can be reduced back to Asc in an NAD(P)H-dependent manner via the corresponding ascorbic acid reductase (AAR). Because NADPH is a primary electron donor for a variety of reduction reactions, such as those catalyzed by glutathione reductase and thioredoxin reductase, the consumption of NADPH increases the degree of cellular sensitivity to oxidative stress. The presence of abundant levels of Asc and free iron may enable the reaction cycle, and the resulting ROS that are produced eventually cause oxidative damage to cells with a reduced redox potential. Researchers in a variety of fields are now intensely interested in iron-dependent, necrotic cell death, a process that is referred to as ferroptosis, in which lipid peroxidation products play a central role in executing cell death [59]. Because HO• promotes lipid peroxidation, ferroptosis may be enhanced by the presence of both Asc and free iron through the pro-oxidant action of this system.

## 5. In vivo Antioxidative Action of Asc Revealed by Animal Studies

Due to the presence of a variety of reductants and oxidants in vivo, it is difficult to clarify the relationship between Asc and specific antioxidative actions in the body [60]. Because primates cannot synthesize Asc, the use of animals that cannot synthesize Asc is advantageous in terms of exploring the physiological roles of Asc. There are several mammalian species among laboratory animals that cannot synthesize Asc, e.g., the guinea pig and a mutant strain of Osteogenic Disorder Shionogi (ODS) rat. The guinea pig has a defect in the GULO gene [61] and has been employed in numerous studies on Asc function in vivo, notably in the cardiovasular system [62]. While rats generally produce Asc, a natural mutant strain, the ODS rat, carries a missense mutatoin in the GULO gene and exhibits growth retardation with the deforming and shortening of the legs, the occurrence of mutiple fractures, oesteo porosis, and hemorrhagic tendencies [63]. However, most studies using these animals have been carried out mainly in Japan where the ODS rat was originally established. Despite some merits of rats as laboratoy animals, this resional limitation may narrow the field of research using the ODS rat compared to genetically modified mice. In this session, after the description on the Asc biosynthetic pathway, we overview findings concerning Asc-deficient mice that have been established by genetic modification.

### 5.1. De novo Synthesis of Asc in Animals

Most animals can still synthesize Asc using Glc as a carbohydrate source via several processes [64]. Figure 5 illustrates the reactions involved in Asc synthesis, with a focus on the last three steps. After being taken up by cells, Glc is phosphorylated to Glc-6-phosphate (Glc-6-P) by the action of hexokinase/glucokinase. While Glc-6-P largely participates in glycolysis, some of the molecules are recruited to the pentose phosphate pathway. A part of Glc-6-P is isomerized to Glc-1-phosphate (Glc-1-P) and then converted to UDP-Glc by glucose-1-phosphate uridylyltransferase. The resulting UDP-Glc is the communal precursor for the synthesis of glycogen, glucuronate, and Asc [64]. For the Asc synthesis pathway, UDP-Glc is oxidized to UDP-glucuronate, and a portion of the UDP-glucuronate may be used for glucuronate conjugation by the action of UDP-glucuronate transferase for the purpose of detoxifying xenobiotic compounds in the liver [65]. The conversion of UDP-glucuronate to D-glucuronate proceeds through either D-glucuronide-1-phosphate or D-β-glucuronide. Two members of the aldo-keto reductase family (AKR), aldehyde reductase (AKR1A) and aldose reductase (AKR1B), are then involved in further reduction of D-glucuronate to L-gulonate [66,67]. The resulting L-gulonate is converted into L-gulono-γ-lactone by the catalytic action of gluconolactonase encoded by RGN, which is identical to the senescence marker protein 30 (SMP30) [68]. L-gulono-γ-lactone in the cytoplasm is then transported into the ER lumen where it is oxidized to Asc by the catalytic action of GULO. Asc is eventually produced by utilizing the oxidizing power of molecular oxygen, while H_2_O_2_ is also produced as a byproduct. A defect (a mutation) in the GULO gene occurred about 63,000,000 years ago in primates [69]. Mutations in GULO have also occurred in some other species, including guinea pigs, bats, Passeriformes birds and teleost fish [70]. These animals are phylogenetically far apart from each other and, hence, appeared to experience the genetic event in the same gene independently [71].

Under hyperglycemic conditions, UDP-Glc is largely utilized for glycogen synthesis by the catalytic action of glycogen synthase. In rodents, either the inhibition of GSH synthesis [72] or the activation of the GSH conjugation reaction [73,74] stimulates Asc synthesis, even though blood Glc levels remain unchanged under these conditions [74]. In this case, Glc-1-P is provided from glycogenolysis by means of the action of glycogen phosphorylase and is converted to UDP-Glc [64]. Hence, an oxidative insult caused by a GSH insufficiency appears to stimulate Asc synthesis irrespective of the glycemic conditions, and fulfill the requirement of Asc in antioxidation in competent animals. Because GSH can recycle DHA to Asc by donating electrons, there appears to be mechanisms that allow these two defense systems to be coordinated against oxidative stress and in maintaining redox homeostasis in the cells.

### 5.2. Phenotypes of GULO-Deficient Mice

There are currently three mice strains that lack either GULO, RGN, or AKR1A in the Asc synthesis pathway and two mice strains that lack the Asc transporters SVCT1 or SVCT2 [10,17,75]. They have been established by genetic modification and are applicable for use in studies on this issue. Because all animals that are naturally incapable of Asc synthesis have a defect in GULO, mice that lack the gene are considered to be ideal models for Asc-incompetent animals, including humans. The GULO^−/−^ mice show a total defect in the ability to synthesize Asc, feeding regular chow containing about 110 mg of Asc/kg does not meet the Asc requirement and causes weight loss, which is rescued by supplementation of 330 mg/liter of Asc in the drinking water [76]. Prolyl hydroxylation in collagen synthesis is considered to be a typical enzymatic reaction that requires Asc. However, no measurable changes in proline hydroxylation or collagen production occurs in the skin, and the degree of angiogenesis associated with tumors or mammary gland growth are also normal in the GULO^−/−^ mouse [77]. These results suggest that another electron donor, most likely GSH, is compensating for the loss of Asc. Analysis of metabolites by proton NMR spectroscopy has revealed differences in metabolism in GULO^−/−^ mice under conditions with and without Asc [78]. There are substantial changes in metabolites, which include compounds associated with antioxidation, notably GSH and possibly uric acid. It should also be noted that the Sfx mouse strain, which is derived from BALB/By mice, with severe combined anemia and thrombocytopenia [79]. The Sfx mouse has a natural deletion in the GULO gene and develops spontaneous fractures at an early age, which are phenotypically similar to those of the GULO^−/−^ mouse [80], although the Sfx mouse is not commonly used for studies.

Asc complements the prolyl hydroxylase that negatively regulates the activity of HIF-1α [3], as described above. While HIF-1α is upregulated in the GULO^−/−^ mouse and blocks neutrophil apoptosis even under normoxic conditions [81], the expression of HIF-dependent genes in the GULO^−/−^ mice is normal, which seems to be achieved by the compensatory effect of GSH on proline hydroxylation in vivo [82]. Apoptosis in spermatocytes is frequently observed in GULO^−/−^ male mice at 20 days of age, which may be caused by a defect in meiosis due to the aberrant expression of heat-shock protein (Hsp) 70 [83]. GULO^−/−^ embryos from heterogeneous parents possess about an equivalent amount of Asc to wild-type (WT) mice, suggesting that sufficient amounts of Asc are delivered from the mother during fetal development [84]. However, on postnatal day 10, the Asc contents in the liver and cerebellum of these mice are markedly low, and the levels of malondialdehyde, an oxidized lipid product, are increased, which supports significance of Asc in antioxidation in these tissues.

It has been suggested that Asc levels are associated with the onset of neurodegenerative diseases. Plasma levels of Asc, but not Toc, are reportedly low in Alzheimer’s disease patients compared to healthy individuals [85]. The APP/PSEN1 mouse, a transgenic mouse that overexpresses both the amyloid precursor protein (APP) and presenilin 1 (PSEN1), is a pathological model for amyloid-β plaque formation, a hallmark and a primary pathogenic indicator of Alzheimer’s disease. GULO^−/−^ mice with APP/PSEN1 transgenes do not show an impairment in spatial learning in an acute Asc deficiency [86]. However, a long-term Asc deficiency leads to hyperactivity and elevated oxidative stress in these mice, suggesting that the cholinergic degradation observed in Alzheimer’s disease may be associated with an Asc deficiency. A decrease in the amounts of dopamine released was reported in GULO^−/−^ mice with APP/PSEN1 transgenes under conditions of an Asc deficiency [87]. There is also another study related to this issue using 5XFAD mice under a GULO^−/−^ background, which is regarded as an early onset transgenic mouse model of Alzheimer’s disease [88]. Indeed, 5XFAD mice carry two APP mutations and two PSEN1 mutations whose expression are regulated by the Thy1 promoter and show amyloid-β pathology. Supplementation of the 5XFAD mice under a GULO^−/−^ background with a higher dose of Asc reduces amyloid plaque and ameliorates integrity of the blood–brain barrier and mitochondrial morphology. A later study using these mice also showed that the Asc status, but not Alzheimer’s disease status, was correlated with oxidative stress in the brain [89].

The highest level of Asc was observed in the cerebellum, olfactory bulbs, and frontal cortex of the Asc-supplemented GULO^−/−^ mice, with the pons and spinal cord having the lowest [90]. Upon supplementation with a low dose of Asc, the level of malondialdehyde still continues to increase in the cortex compared to the cerebellum and pons, suggesting that the cortex requires more Asc for protecting against oxidative stress than other parts of brain. An Asc deficiency during gestation induces intraparenchymal hemorrhages and defects in the development of the cerebellum [91]. Elevated oxidative stress in the cortex and cerebellum, a decreased strength and an agility deficit are observed in the adult GULO^−/−^ mice that are supplemented with low levels of Asc [92]. Moderate levels of Asc appear to be required for normal neuronal function because GULO^−/−^ mice that are supplemented with low levels of Asc (220 ppm) show mild movement disorders, but also show an exaggerated hyperactivity to dopamine agonists [93]. Insufficient supplementation of Asc to GULO^−/−^ mice causes a decrease in voluntary locomotor activity, diminished physical strength, and an increased preference for a palatable sucrose reward during scorbutic periods [94]. This aberrant behavior appears to be associated with decreased blood Glc levels, elevated oxidative damage in the cortex, or a decrease in the levels of metabolites derived from dopamine and serotonin, which are observed under an Asc deficiency. With Asc supplementation, these abnormal behaviors revert to those of the control mice. Without Asc supplementation, motor performance decreases in the GULO^−/−^/SVCT2^+/−^ mice, while a Toc deficiency alone does not cause such an aberrant phenotype [95].

An Asc insufficiency also dominantly affects the cardiovascular system. The development of atherosclerosis is enhanced in mice with a double deficiency of GULO and ApoE [96]. A reduction in the amount of collagen produced is observed in advanced atherosclerotic plaques in the GULO^−/−^ mice. The mice also show large necrotic cores within plaques, and a reduced fibroproliferation and neovascularization in the aortic adventitia, which may increase vulnerability to rupture. Normal vascular function may also be impaired by low levels of Asc. An Asc deficiency accelerates the proteasomal degradation of aldehyde dehydrogenase-2 (ALDH2), which catalyzes the bioactivation of nitroglycerin [97]. Accordingly, GULO^−/−^ mice are tolerant to nitroglycerin-induced vascular relaxation. An Asc deficiency in GULO^−/−^ mice increases the levels of both low-density lipoprotein (LDL) and proprotein convertase subtilisin/kexin 9 (PCSK9), which suppress clearance from the circulation by stimulating the degradation of LDL receptors [98]. These collective data suggest that Asc induces the formation of LDL receptors through the activation of SREBP2, but, PCSK9 is suppressed through the activation of Fox32a. Water restrain stress accelerates the death of GULO^−/−^ mice as a consequence of cardiac damage [99]. An Asc insufficiency results in the increased death of cardiomyocytes, which appears to be associated with increases in the expression of matrix metalloprotease (MMP)-2 and -9 and lipid peroxidation products.

An Asc deficiency in GULO^−/−^ mice aggravates the lung pathology caused by an influenza virus infection [100]. Because Asc supplementation alleviates the pathological conditions in the lung, Asc plays a role in the development of an adequate immune response against the influenza virus in the lungs. In the meantime, an Asc deficiency inhibits Cl^−^ secretion in the airway epithelium, which appears to be caused by a decreased expression of the cystic fibrosis conductance regulator (CFTR) [101]. Natural immunity that includes the secretion of IFN-γ, the expression of perforin and granzyme B, and cytotoxic activity against ovarian cancer cells is reduced in natural killer (NK) cells in GULO^−/−^ mice [102]. Although an infection with *Klebsiella pneumoniae* renders the GULO^−/−^ mouse vulnerable and lethal [103], Asc supplementation does not affect levels of F_2_-isoprostan, a marker for lipid peroxidation, or oxidized amino acids. Thus, Asc appears to have a previously unappreciated role in host defense mechanisms against invading pathogens. While the expression of the viral-sensing receptors, retinoic acid-inducible gene 1 (RIG-1), and the melanoma differentiation-associated protein 5 (MDA-5), are enhanced by Asc in the human bronchial epithelium transformed with Ad12-SV40 2B (BEAS-2B) cells, the expression of these genes are markedly decreased in the lungs of the GULO^−/−^ mouse [104].

Treatment of GULO^−/−^ mice with 3,3′,4,4′,5-pentachlorpheny induces severe porphyria cutanea tarda, and Asc effectively suppresses the accumulation of uroporphyrin in the liver [105]. The administration of excessive levels of iron, however, invalidates the suppressive effects of Asc. The administration of concanavalin A induces liver injury in GULO^−/−^ mice, as evidenced by an increased hepatocyte apoptosis and severely elevated levels of the pro-inflammatory cytokines, TNF-α and IFN-γ under an Asc insufficiency [106]. While the levels of interleukin-22, a hepatoprotective cytokine, are high, there are defects in the expression of the receptor IL-22Rα and hence downstream STAT3 signaling is not activated. The administration of thioacetamide, a hepatotoxin, to GULO^−/−^ mice causes a decreased survival rate, aggravation of liver damage, and enhanced fibrosis, which are less evident in Asc-supplemented mice [107]. While lithocholic acid induces liver damage and hepatic fibrosis in GULO^−/−^ mice more severely than in WT mice, Asc attenuates these issues in association with decreased apoptosis and necrosis [108]. Since an Asc insufficiency results in an increase in the levels of oxidative stress markers in the thioacetamide-treated mice, the antioxidative action of Asc would be involved in liver protection. Investigations of the IRE1α and eIF2α proteins, which are involved in the ER homeostasis, imply that Asc supplementation alleviates the ER stress that occurred spontaneously in the livers of GULO^−/−^ mice [109].

While Asc-deficient GULO^−/−^ mice are more susceptible to the multiple organ dysfunction syndrome, which can be the cause of death by severe sepsis, an infusion of Asc effectively attenuates these pathological conditions [110]. The neutrophil extracellular trap (NET), which is a novel mechanism for killing pathogens, is produced under conditions of sepsis and is substantial in Asc-deficient GULO^−/−^ mice [111]. The formation of excess amounts of NET may injure tissues under pathogenic conditions, such as sepsis, but, in the Asc-supplemented GULO^−/−^ mouse, becomes attenuated. This beneficial action of Asc may be explained by the suppression of the mitogen-activated protein kinase (MAPK) and NF-κB signaling. An Asc deficiency aggravates TNFα-induced insulin resistance and stimulates lipid accumulation and inflammation in the liver [112] and a diabetic glomerular injury induced by the administration of streptozotocin to GULO^−/−^ mice [113].

The growth and metastasis of B16FO murine melanoma cells is increased in the GULO^−/−^ mice [114]. Asc administration decreases the levels of serum inflammatory cytokines IL-6 and IL-1β and concomitantly decreases tumor size. Similarly, a reduction in tumor weight and serum inflammatory cytokine IL-6 levels were observed in 4T1 breast cancer cells that had been transplanted to Asc-supplemented GULO^−/−^ mice [115]. GULO^−/−^ mice are more susceptible to the carcinogenic effects of nickel subsulfide (Ni_3_S_2_), whereas supplementation with Asc tends to attenuate the acute toxicity of Ni_3_S_2_ and to extend the latency of transplanted tumors [116]. Pharmacokinetic analyses of i.p. administered high doses of Asc to GULO^−/−^ mice indicate that the elimination of Asc from solid tumors (Lewi lung tumors) is markedly slower than that from plasma and the liver (t_1/2_~80 min) [117]. Thus, Asc alleviates both tumorigenesis and tumor development in GULO^−/−^ mice.

### 5.3. Phenotypes of RGN/SMP30-Deficient Mice

From an enzymatic point of view, the activity of RGN that catalyzes the conversion of L-gulonic acid to L-glono-γ-lactone had been known for some time, but the issue of which gene product that actually performed the reaction remained ambiguous. An analysis of SMP30-knockout mice eventually identified the SMP30 protein as the RGN gene product [68]. The abbreviation SMP30 is still commonly used, but the actual name for the gene, RGN, is used throughout this article. The expression of RGN decreases with the aging of major tissues, including the heart, lung, liver, and kidney [118]. RGN^−/−^ mice show a lower body weight and a shorter life span than WT mice, but they are viable and fertile as long as they receive Asc supplementation [119]. RGN^−/−^ mice without Asc supplementation show an increase in ROS formation and accelerated senescence in the tubular epithelia in the kidney [120]. Asc supplementation suppresses the production of protein carbonyl groups, which are representative markers of oxidative stress, but has no effect on lipid peroxidation products, another hallmark of oxidative stress, in RGN^−/−^ mice [121].

The use of RGN^−/−^ mice provided information concerning the distribution of Asc in the body after ingestion. Asc is distributed to the liver within 3 h, but longer times (~12 h) are needed for some organs, such as the central nervous systems, the testes, and the thyroid gland [122]. Consistent with the reported actions of Asc, RGN^−/−^ mice show decreased levels of adrenalin and noradrenalin in adrenal glands [123]. The collagen content in the lung is decreased in RGN^−/−^ mice, but there are no changes in hydroxyproline contents in the skin, despite morphological abnormalities in the epidermis [124]. Transcriptome analyses of the skin of RGN^−/−^ mice show the altered expression of some genes, notably those associated with hair growth [125]. Asc is believed to be required for the biosynthesis of carnitine but appears to not be essential for its synthesis again [126].

Muscular atrophy and deteriorated physical activity are observed in RGN^−/−^ mice [127]. Dual-energy X-ray absorptiometry data indicate that the bones are rough and porous with low mineral content and mineral density in RGN^−/−^ mice, which can be largely rescued by Asc administration [128]. Tissue damage, including abnormal cardiac dilation and cognition of the liver and lungs, are observed in the fetus and neonate RGN^−/−^ mice delivered from mothers without Asc supplementation [129]. The hypermethylation of DNA observed in the RGN^−/−^ mouse liver is restored to the demethylation state by the administration of a high dose of Asc at the stage of lactation [130], which is consistent with Asc having a role in the regulation of DNA methylation status [29].

ROS generation and NADPH oxidase activities are elevated, while expression of the major antioxidative enzymes remains unaffected in the brains of RGN^−/−^ mice [131]. An increase in the auditory brainstem response thresholds and a decrease in the number of spiral ganglion cells suggest that Asc-deficient RGN^−/−^ mice suffer more severe age-related hearing loss [132]. Hallmarks of somatic and visceral pain sensitivity are enhanced in the RGN^−/−^ mice, suggesting that Asc has a role in the regulation of pain sensitivity [133]. Asc status may also play a critical role in anxiety and anorexia in these mice [134]. Mice with ablation of both RGN and the α-Toc transfer protein show normal locomotor activity and higher levels of anxiety, but their fear memory is impaired, compared with the mice with sufficient Asc and α-Toc [135]. Thus, Asc insufficiency is thought to affect higher brain function, consistent with the high levels of Asc in brain tissue in WT mice. Hydrogen-rich pure water prevents elevated O_2_•^−^ formation in brain slices of RGN^−/−^ mice [136], which suggests a compensatory function of hydrogen to Asc in elimination of O_2_•^−^.

An increased release of angiotensin and H_2_O_2_ are observed in cardiomyocytes isolated from RGN^−/−^ mice upon oxidative stress [137]. The cardiotoxic effects of doxorubicin, an anti-cancer agent that triggers the production of ROS, are enhanced in RGN^−/−^ mice [138]. Oxidative stress due to an Asc deficiency results in thiol oxidation and endothelial dysfunction in RGN^−/−^ mice, leading to coronary artery spasms [139]. Ischemia-reperfusion injury is exacerbated irrespective of Asc status [140], but collateral growth decreases due to impaired angiogenesis in the hearts of RGN^−/−^ mice [141]. Because Asc-sufficient RGN^−/−^ mice still show an exacerbation of angiotensin II-induced cardiac hypertrophy and remodeling, RGN appears to exert its cardioprotective effects independent from the Asc production [142,143].

Elevated oxidative stress and a decreased level of collagen synthesis occur in RGN^−/−^ mice at 1–3 months of age, which appear to be responsible for pulmonary emphysema due to an Asc deficiency [144]. Asc administration suppresses ROS production in the lung [145] and prevents aggravated smoke-induced emphysema in RGN^−/−^ mice, suggesting a potential application of Asc as a therapeutic agent for the treatment of chronic obstructive pulmonary disease (COPD) [146]. However, it is noteworthy that excess Asc increases the air space size and lipid peroxidation products, which may be attributed to the pro-oxidant action of Asc in the lung [144]. As was mentioned for brain slices [136], hydrogen-rich pure water also prevents cigarette smoke-induced pulmonary emphysema in RGN^−/−^ mice [147].

The use of RGN^−/−^ mice unveils the potential roles of RGN in Glc metabolism and non-alcoholic fatty liver diseases, as was recently overviewed [148]. Impaired Glc tolerance and lower blood insulin levels are characteristically observed in RGN^−/−^ mice [149]. The expression of the transporters for Asc (SVCT1 and SVCT2) and DHA (GLUT1, GLUT3, and GLUT4) are upregulated in the livers of RGN^−/−^ mice [150], most likely due to a compensatory reaction in response to an Asc insufficiency. Accordingly, primary hepatocytes isolated from RGN^−/−^ mice take up more Asc from the extracellular space. Triglycerides, cholesterol, and phospholipids accumulate in the livers of RGN^−/−^ mice, even when they are fed a conventional diet [151], which results in modest Glc tolerance with an impaired insulin secretion in the early developmental stage [152]. However, the authors conclude that the disorder in Glc metabolism is not associated with an Asc deficiency but, rather, is caused by other mechanism attributable to the ablation of RGN.

A microarray analysis of RGN^−/−^ livers revealed an increased expression of some genes, which are involved in redox reactions under the regulation of Nrf2 and lipid metabolism [153]. An Asc deficiency in the RGN^−/−^ mice elevates the level of oxidative stress, which results in the activation of SREBPs in hepatocytes via the stimulation of ER stress responses. Accordingly, the double deficiency of RGN and the leptin receptor increases LDL, the development of a severe fatty liver, and an increased ER stress that are not evident in RGN^−/−^ mice with Asc supplementation [154]. A recent study, however, indicates that a long-term Asc deficiency in RGN^−/−^ mice rather inhibits lipogenesis through the impairment in SREBP1c activation [155]. The accumulation of excessive levels of cholesterol in RGN^−/−^ hepatocytes appears to be responsible for this impotency of SREBP1c in this case. Asc dramatically attenuates ethanol-mediated liver injury in ethanol-fed RGN^−/−^ mice by suppressing the infiltration of neutrophils which is accompanied by less CD68-positive cell infiltration [156]. Thus, Asc appears to play pivotal roles in lipid homeostasis and the development of nonalcoholic liver disease (NAFLD), although further studies will be required to reveal the complete picture.

An elevation in ROS caused by an Asc deficiency induces the expression of a member of cytochrome P450 (CYP) proteins, e.g., CYP2E1, which stimulates oxidative damage in hepatocytes of RGN^−/−^ mice and appears to be the underlying mechanism for the binucleation observed in such mice [157]. Treating RGN^−/−^ mice with carbon tetrachloride (CCl_4_) causes severe liver injury, but this injury is protected by arazyme, a protease produced by a gram-negative aerobic bacterium (*Aranicola proteolyticus* HY-3 strain) [158]. Contrary to the antioxidative action of Asc, supplementation with Asc aggravates CCl_4_-induced hepatic injury in the RGN^−/−^ mice [159]. Liver fibrosis caused by treatment with CCl_4_ is attenuated in Asc-deficient RGN^−/−^ mice compared with WT mice and is reversed by the administration of Asc [160]. Because the peroxisome proliferator-activated receptor-γ (PPAR-γ) is upregulated in Asc-deficient RGN^−/−^ mice, suppression in the liver fibrosis may be associated with the genes that are regulated by PPAR-γ.

RGN^−/−^ mice are more susceptible to inducers of apoptosis, i.e., TNF-α and Fas, compared to WT mice [119]. RGN appears to regulate NF-κB through balancing the activities of protein kinase and protein tyrosine phosphatase, so that the age-related decrease in RGN protein per se causes NF-κB activation and exacerbation of the inflammatory process [161]. However, the issue of whether or not Asc is involved in this process was not examined in this study. While aged WT mice show attenuation in the phagocytotic removal of secondary necrotic cells, Asc-deficient RGN^−/−^ mice, even at an early stage, show a similar defect [162]. However, the inflammatory responses that are enhanced by secondary necrotic neutrophils are similar between RGN^−/−^ mice and aged WT mice [163]. The administration of dexamethasone, a synthetic glucocorticoid, decreases CD4^+^ and CD8^+^ T cells in splenocytes of RGN^−/−^ mice. This immune-suppressive action of dexamethasone is mitigated by the intake of Asc [164]. Because Asc supplementation increases GSH and SOD, the authors conclude that maintaining antioxidative activity exerts protection against glucocorticoid toxicity. While colitis induced by the ingestion of 3% (w/v) dextran sodium sulfate is aggravated in RGN^−/−^ mice, Asc has no ameliorating effects [165].

The upregulation of pro-inflammatory genes is observed when the kidney is subjected to γ-ray irradiation, which appears to be caused by the activation of NF-κB in Asc-deficient RGN^−/−^ mice [166]. Because the overexpression of RGN in the Smad3^−/−^ mouse suppresses γ-ray irradiation-induced apoptosis [167], Asc appears to suppress inflammatory responses through mitigating the activation of NF-κB that is stimulated by γ-ray irradiation. Similarly, γ-irradiation aggravates intestinal injury in RGN^−/−^ mice compared to WT mice, which is also protected by Asc supplementation [168]. In the meantime, Asc supplementation fails to protect against γ-irradiation-induced apoptosis in the crypts of the small intestine in the RGN^−/−^ mice [169]. Thus, further studies will clearly be required to understand the exact mechanism of how Asc protects against irradiation-induced intestinal injury. An analysis of RGN expression by means of immunohistochemistry suggests that a low expression of RGN is associated with the development of non-small cell lung cancer [170]. In vitro experiments using a cell line suggests that RGN exhibits anti-tumor effects through the suppression of histone deacetylase 4 (HDAC4). When diethylnitrosamine, a popular carcinogenic compound, is administered to RGN^−/−^ mice, severe hepatic damage is observed, and, due to the high toxicity of this compound, all of the mice died before developing cancer [171].

The administration of the β-adrenergic stimulant, isoproterenol, causes little change in secretory granules in granular duct cells in RGN^−/−^ mice compared to WT mice, which suggests that the β-adrenergic signal is modulated by RGN [172]. In diabetic nephropathy induced by treatment with streptozotocin, injury in the proximal tubule is severe in RGN^−/−^ mice and supplementing Asc has no effect on this [173]. The double deficiency of RGN and SOD1 causes premature death [174]. The authors suggest that a malfunction in plasma lipid metabolism and the accumulation of hepatic lipids due to elevated oxidative stress are the potential cause for this death. However, mice with a double deficiency of AKR1A, another gene responsible for Asc synthesis, and SOD1 die due to the development of acute pneumonia after Asc administration is stopped [175] as described below. Thus, this issue needs to be re-examined from a pathological standpoint. Asc is present at quite high levels in the lens in healthy mice, and UV-induced cataract formation is more severe in Asc-deficient RGN^−/−^ mice than WT-mice [176]. These results suggest that a major function of Asc in the lens is protection against UV irradiation, likely via its antioxidative function. Although the responsible mechanism has not been clarified, the overexpression of RGN in HepG2 cells, which are human hepatoma-derived cells lacking GULO and hence unable to synthesize Asc, results in a decrease in the levels of ROS and lipid peroxidation products [177]. RGN appears to have an unknown function that is independent from the Asc synthesis but is associated with antioxidation. In fact, RGN hydrolyzes diisopropyl phosphorofluoridate [178], which suggests that there are other compounds that also serve as substrates for RGN activity. Investigations from the standpoint of enzymology may provide answers to this issue.

### 5.4. Phenotypes of AKR1A-Deficient Mice

The AKR superfamily of enzymes catalyze the NADPH-dependent reduction of a wide range of aldehydes to their corresponding alcohols. AKR1A and aldose reductase (AKR1B) are close members in the AKR superfamily and exhibit overlapping activities for many substrates that include D-glucuronate, which is involved in the synthesis of Asc and a variety of aldehydes, such as methylglyoxal and 3-deoxyglucosone, the levels of which are elevated under conditions of oxidative stress, inflammation, and hyperglycemia [179]. However, while AKR1B uniquely catalyzes the conversion of Glc to sorbitol and, along with sorbitol dehydrogenase, constitutes the polyol pathway, which is associated with the development of diabetic complications, AKR1A does not.

There are several processes that require redox reactions in the synthesis of Asc from Glc, and the involvement of AKR enzymes in the reduction of D-glucuronate to L-gulonate had been expected [64]. The actual participation of AKR1A and AKR1B in Asc synthesis was demonstrated by the characterization of AKR1A^−/ −^ mice and AKR1B^−/ −^ mice [66,67]. AKR1A and AKR1B contribute to 85–90% and 10–15% of the total Asc synthesis, respectively, in mice. The difference in their contribution is not likely due to their enzymatic properties but, rather, to differences in their tissue distribution. The liver largely expresses AKR1A and is the central organ for Asc synthesis. Due to the lower expression of AKR1B compared to AKR1A and, hence, a minor contribution to Asc synthesis in the liver, the contribution of AKR1B in the synthesis of Asc had not been recognized beforehand. Consistently, no phenotypic changes attributable to Asc function have been reported in AKR1B^−/−^ mice [66]. Phenotypic abnormalities derived from Asc in AKR1A^−/ −^ mice are largely similar to those for GULO^−/−^ and RGN^−/−^ mice, but there are some differences, seemingly based on the unique features of AKR1A from other genes. In some cases, phenotypic changes caused by an AKR1A deficiency are completely independent from the Asc status [180,181]. Regarding the roles of Asc in the hydroxylation of proline residues in collagen, a pathological model for kidney fibrosis caused by unilateral ureteral obstruction showed no evident difference in collagen levels between the AKR1A^−/−^ and WT mice [182]. Such observations on collagen are consistent with those in the skin of GULO^−/−^ mice [77] and RGN^−/−^ mice [124].

Bone morphogenesis is severely affected by the ablation of AKR1A [66]. Mice in which eGFP is knocked-in on the locus of AKR1A4, a mouse orthologue of AKR1A, may be useful for investigating the osteoporosis caused by the ablation of AKR1A ablation [183]. Analyses of the developmental divergence in the size of AKR1A^−/−^ neonatal mice indicate an inverse correlation between plasma Asc levels and corticosterone levels [184]. These high levels of corticosterone are sustained in adulthood even by Asc supplementation and appear to be responsible for the aggressive behavior of AKR1A^−/−^ mice [180]. The administration of pentobarbital causes stronger anesthetic effects, which are characterized by a prolonged sleep time, in AKR1A^−/−^ mice compared to WT mice [185]. Because receptors for γ-aminobutyric acid (GABA) are thought to be the likely target of pentobarbital, the influence of pentobarbital on GABA receptors may be sustained under an Asc insufficiency. Impaired spatial memory formation in juvenile AKR1A^−/−^ mice was observed based on a water maze test, although there is no noticeable hippocampal damage or neurotransmitter contents [186].

A double deficiency of AKR1A and SOD1 inevitably causes premature death in the mice within two weeks after the cessation of Asc supplementation [175]. The life span of the double deficient mice is significantly extended by the administration of edaravone, a radical-scavenging compound. Thus, oxidative lung injury is the primary cause of death of the double deficient mice. Because the lung is an organ that is continuously exposed to hyperoxygenic conditions, oxidative damage due to an Asc deficiency emerges most prominently in the lung and causes acute pneumonia. Precise analyses of their lungs may provide clues to the pathogenesis of acute respiratory distress syndrome (ARDS) in which the oxidative damage of pulmonary tissues is deeply involved [187].

The livers of AKR1A^−/−^ mice are significantly heavier than those from AKR1A^+/+^ mice [185], which may be associated with altered glycogen metabolism [188] and differential responses to hepatotoxic compounds in AKR1A^−/−^ mice [189,190]. Treatment with CCl_4_ causes severe hepatic damage with enhanced steatosis in the livers of AKR1A^−/−^ mice compared to WT mice, and Asc supplementation reverses these changes, making them comparable to those in WT mice [189]. The aggravation in hepatic damage is consistent with observations regarding RGN^−/−^ mice [158]. However, there are contradictions in the actions of supplemented Asc between AKR1A^−/−^ mice and RGN^−/−^ mice; while Asc is protective for AKR1A^−/−^ mice [189], it is detrimental for RGN^−/−^ mice [158]. These seemingly inconsistent observations might be due to differences in the unique actions of these genes, so clarification is also awaited regarding this issue as well. Hepatic damage caused by an overdose of acetaminophen induces more severe damage in AKR1A^−/−^ mice compared to WT mice, and the hepatic damage is effectively rescued by Asc supplementation [191,192]. We now know that this aggravation in hepatic damage in AKR1A^−/−^ mice is associated with the coordinated elevation of three conjugation reactions of acetaminophen, which consist of glucuronidation, sulfation, and glutathione conjugation [193]. Based on these observations, it is conceivable that the consumption of GSH and ATP by such stimulated conjugation reactions coordinately lead to the aggravated liver damage that is observed under conditions of an Asc insufficiency. Surprisingly, however, AKR1A^−/−^ mice are resistant to treatment with thioacetamide under Asc-insufficient conditions compared to WT mice [194]. Although both acetaminophen and thioacetamide experience metabolic activation by CYP, corresponding CYP members differ for these compounds. Accordingly, the inconsistency in observations between the two hepatotoxicants may explain the differential responses of CYP in the AKR1A^−/−^ mice. Aberrant function of CYP is also known as a cause of chemical carcinogenesis of liver. Diethylnitrosamine, a tumorigenic nitroso compound activated by CYP, is highly toxic to AKR1A^−/−^ mice and results in an extreme fatality rate [190]. A few AKR1A^−/−^ mice that survived after a half-year treatment with diethylnitrosamine developed numerous tumor nodules in the liver. A marked alleviation was observed in the case of mice with Asc supplementation, which supports the beneficial action of Asc in suppressing tumor development.

### 5.5. Phenotypes of SVCT-Deficient Mice

Asc is either delivered to the liver via the hepatic portal vein after adsorption in the intestine or synthesis in the liver of competent animals. When transporters do not properly function, many organs experience an Asc deficiency even in the competent animals. While the intestinal absorption of Asc has been reported as a role of SVCT1, intestines of mice with SVCT1 gene ablation show a normal Asc status [14]. This may be explained by compensatory uptake of DHA via GLUT2 and GLUT8. On the other hand, SVCT2 is responsible for the tissue accumulation of Asc, which was confirmed by observations of phenotypes of SVCT2^−/−^ mice [195]. Asc has essential roles in the brain for supporting certain types of enzymatic reactions, such as catecholamine synthesis, and antioxidation [15,20]. GLUT1, which is expressed on endothelial cells at the blood-brain barrier, may transport DHA into the brain and supply Asc to neurons [196]. A high expression of SVCT2 was observed in the cortex and cerebellum of fetal rat brains [197]. Consistent with the abundant presence, SVCT2-decifient mice show intraparenchymal brain hemorrhages and die from respiratory failure within a few minutes after birth [195]. The functions of adrenal chromaffin cells are also impaired, which results in decreases in the tissue adrenalin/noradrenalin and plasma levels of corticosterone [198]. While Asc is abundantly present in the ER where GULO catalyzes the final process of its synthesis, mitochondria are also rich in Asc, which may be explained by the localization of SVCT2 in the mitochondrial membrane [199]. Some GLUTs are also responsible for the uptake of DHA, so it is rather difficult to understand the unique function of an individual transporter from the standpoint of the physiological actions of Asc/DHA by means of the knockout of a single gene.

## 6. Asc in the Prevention and Treatment of Cancer

Potential applications of Asc for the treatment of cancer was first proposed by Linus Pauling who showed that the administration of high doses of Asc to cancer patients had a beneficial action [200]. Subsequent studies, however, failed to confirm the advantage of this in cancer treatment [201], and the validity of using Asc for this purpose has been the subject of much debate [202]. Advances in cancer research from the aspects of basic biology and clinical applications are now revealing the advantages of both antioxidative action and pro-oxidant action of Asc and DHA in the prevention of tumorigenesis and in the treatment of cancer patients [9,202].

### 6.1. Suppression of Carcinogenesis by Genetic and Epi-genetic Actions of Asc

Stimuli that modify or destroy DNA integrity cause mutations in genes that may lead to carcinogenesis. ROS, most notably HO•, preferentially oxidizes purine bases of nucleotides, especially the carbon at position 8 of guanine, which results in the formation of an equilibrium mixture of 8-hydroxyguanine (8-OHdG) and 8-oxo-guanine [203]. The administration of Asc in the form of DHA to cultured cells has been shown to prevent H_2_O_2_-induced base substitution on the transfected chloramphenicol acetyl transferase gene [204]. Surveys of human studies indicate that Asc appears to be protective against oxidative DNA modification as well as chromosomal abnormalities at a nearly physiological concentration range above 50 µM [205].

While ROS are involved in the mutagenesis of genes, impaired redox reactions may also affect several processes of carcinogenesis and cancer development caused by oxidative stress [206]. When the effects of Asc and a lipophilic derivative, 2-O-octadecylascorbic acid, were examined in rats that develop carcinogenesis by feeding a choline-deficient, L-amino acid-defined diet, a lipophilic Asc derivative appears to prevent cancer development more effectively than Asc [207]. The anti-tumorigenic effects of Asc can also be examined in model animals with chemical-induced carcinogenesis. Nitrosamines are oxidatively activated by CYP, such as CYP2E1, and then become hepato-carcinogens. Asc reportedly suppresses the formation of 8-OHdG in *N*-nitrosodimethylamine-treated cells [208]. It is also noteworthy that the anti-tumor action of an Asc derivative, 5,6-benzylidene-L-ascorbate, in *N*-nitrosodiethylamine-treated mice is reportedly mediated by the pro-oxidant action of this molecule [209].

Moreover, the epigenetic actions of Asc on both DNA and histones also appear to be involved in the prevention of tumorigenesis [4,5]. Asc may exert anti-carcinogenic activity through electron-donation to TET, which initiates the replacement of 5mC with unmodified cytosine on DNA [210]. As a result, Asc would be able to support the reactivation of genes, including anti-tumor genes, that had been epigenetically silenced by the methylation of cytosine residues that are dominantly present in the CpG island in the promoter region [4]. Moreover, because the excessive methylation in lysine residues of histones solidify the chromatin structure, demethylation via the Asc-involved reaction also aids in the transcriptional activation of epigenetically silenced anti-tumor genes [33]. Thus, Asc appears to be advantageous in preventing tumorigenesis from both genetic and epi-genetic aspects.

Sulfhydryl groups in some enzymes are highly sensitive to oxidative insult and can be inactivated transiently or permanently, depending on the extent of oxidative stress. For example, the oxidative inactivation of phosphotyrosine phosphatase causes the sustained activation of signaling of the growth factor receptor [46]. Accordingly, excessive ROS may cause uncontrolled cell proliferation and lead to an increase in risk for cellular transformation [47]. In fact, the inactivation of some phosphatases that are responsible for receptor signaling are associated with tumors in a variety of tissues [211]. For example, either a mutation or hyperoxidation causes the inactivation of phosphatase and a tensin homolog that was deleted from chromosome 10 (PTEN), which leads to the acceleration in Akt-mediated tumor development [212]. Because Asc acts as a general antioxidant, the intake of sufficient amounts of Asc would eventually suppress the risks for tumorigenesis by eliminating ROS or reducing the Cys-SOH in enzymes that control cell growth.

### 6.2. Therapeutic Use of Asc for Cancer Treatment

While physiological levels of Asc may suppress tumorigenesis partly by suppressing the production of ROS, pharmacological doses of Asc may exert anti-cancer effects via its pro-oxidant action [52,213]. Asc levels in the blood plasma are strictly controlled through several mechanisms, so that the oral administration of a high dose of Asc does not reach pharmacologically effective levels. For purposes of cancer therapy, a high dose of Asc is generally administered via intravenous injection. The constitutive activation of HIF-1α induces genes that are involved in the supply of oxygen and energy production, leading to the development of many solid tumors being stimulated [214]. It is likely that the presence of physiological levels of Asc control HIF-1α to appropriate levels through the proline hydroxylation associated with pVHL-mediated proteasomal degradation. A large excessive amount of Asc together with the presence of oxygen may restrict the proliferation and survival of tumor cells [3], and hence, pharmacologically high doses of Asc would act as a potential therapeutic agent for some types of cancers [215]. For example, the administration of 10 g of Asc results in a 1–5 mM concentration in the blood plasma of cancer patients [216]. SVCT2 appears to be responsible for the uptake of high doses of Asc and is involved in the induction of cell death in breast cancer [217] and cholangiocarcinomas [218].

The oxidative metabolism of carbon compounds in mitochondria is stimulated by Asc treatment and produces O_2_•^−^ and H_2_O_2_, which disrupts iron metabolism in cancer cells [219]. Ras mutations have been reported in multiple diethylnitrosamine-induced HCC specimens [220]. Cancer cells become sensitized to ROS produced by interactions between labile iron and H_2_O_2_ in some types of cancer cells, notably those bearing tumorigenic mutations in KRAS and BRAF [221]. In this case, however, DHA mediates the oxidative insult that selectively occurs in cancer cells bearing mutations in KRAS and BRAF. An antibody directed against the epidermal growth factor receptor, cetuximab, has been reported to be an effective therapy for patients with WT KRAS but not with mutant KRAS. The expression of SVCT2 appears to determine the therapeutic potential of Asc in cetuximab-resistant colon cancer patients with mutant KRAS [222]. The administration of Asc to cancer patients with a fasting-mimicking diet appears to be effective therapeutics for the treatment of colorectal cancer and other tumors with KRAS mutations [223]. These anti-cancer effects are associated with ROS production during the metabolism of DHA; the reduction of DHA to Asc is associated with the consumption of GSH, which results in an elevation in ROS levels and oxidative DNA damage due to dysfunctional glutathione peroxidase. The activation of the poly-(ADP-ribose) polymerase then results in the consumption of NAD^+^, which consequently depletes energy, thus causing eventual cancer cell death. However, a survey of the literature on both human and animal studies has not reached a consensus on whether Asc has a beneficial effect in cancer treatment or not [224], and further studies may be required to clarify this issue.

Immune systems that suppress tumorigenesis are also regulated by Asc [225]. For example, a high dose Asc may demethylate DNA and potentiate anti-PD1 checkpoint inhibition through a synergistic mechanism [226]. Another example is that Asc and its derivative ascorbic acid 2-phosphate have been shown to modulate the expansion of and cytokine production in human γδ T cells [227]. Thus, a pharmacologic dose of Asc, in combination with immunotherapies, may be more effective in the treatment of some types of cancer. In general, Asc is being confirmed to have anti-cancer effects, which is a notion originally proposed by Pauling [198]. Because multiple factors are involved in carcinogenesis and Asc is a pluripotent agent, further studies will be needed to reveal the complete picture regarding this issue.

## 7. Future Perspectives

Asc, as an essential micronutrient, is involved in a variety of redox reactions, which include enzymatic reactions that require redox potential and non-enzymatic radical-scavenging reactions. While the antioxidative ability of Asc suppresses the pathogenesis of diseases related to the development of oxidative stress as typically occurs under inflammation, redox cycling mediated by the presence of free iron may cause Asc to act as a pro-oxidant as well, which can be employed for tumoricidal treatment. In addition to animals with natural mutations in GULO, genetically modified mice that are incapable of synthesizing or transporting Asc indicate the pathological conditions that are associated with an Asc deficiency, including oxidative damage. By employing these mice, we could obtain new knowledge that was not obtained from studies on in vitro chemical reactions or cultured cells. For example, regarding the in vivo action of Asc as a cofactor for enzymatic reactions, the contribution of complementary molecules that act as electron donors has emerged. On the other hand, the usefulness of Asc as an antioxidant has been found to be more important than previously assumed. However, the use of genetically modified mice faces several challenges to overcome. The mouse metabolism is significantly different from humans and, hence, caution should be taken in extrapolating the results obtained from mice to understanding human physiology or pathology. There are also limitations to the surgical procedures that can be performed when creating disease models in mice. Because rats are approximately 8–10 times larger and exhibit more similarity to human physiology than mice, they have been emplyed in a variety of studies. The application of the genome editing technique to other laboratory animals, which exhibit a higher similarity to humans, could reveal details concerning the physiological and pharmacological actions of Asc that include the central nervous system, the cardiovascular system, and cancer biology and treatment.

## Figures and Tables

**Figure 1 molecules-27-06187-f001:**
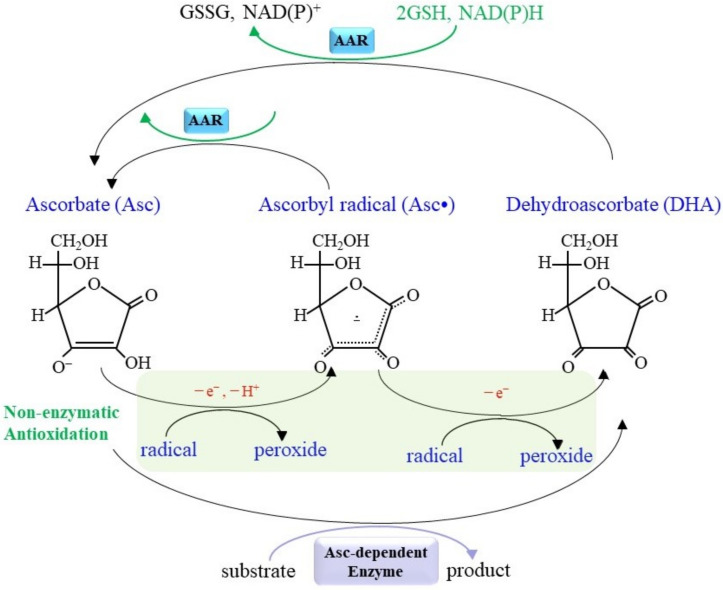
Schemes for the oxidation of Asc and reductive recycling of Asc• and DHA to Asc. Three major forms of ascorbic acid and their redox-mediated interconversion are depicted. Reaction with a radical species causes the one electron oxidation of Asc and results in the production of Asc•. Either dismutation between two Asc or interaction with another radical species may lead to formation of DHA. Enzymatic reactions that utilize Asc as an electron source generally cause a two-electron oxidation and result in DHA formation. Asc• and DHA can be reduced back by either AAR using NAD(P)H or non-enzymatic reactions with GSH or an electron from ETC.

**Figure 2 molecules-27-06187-f002:**
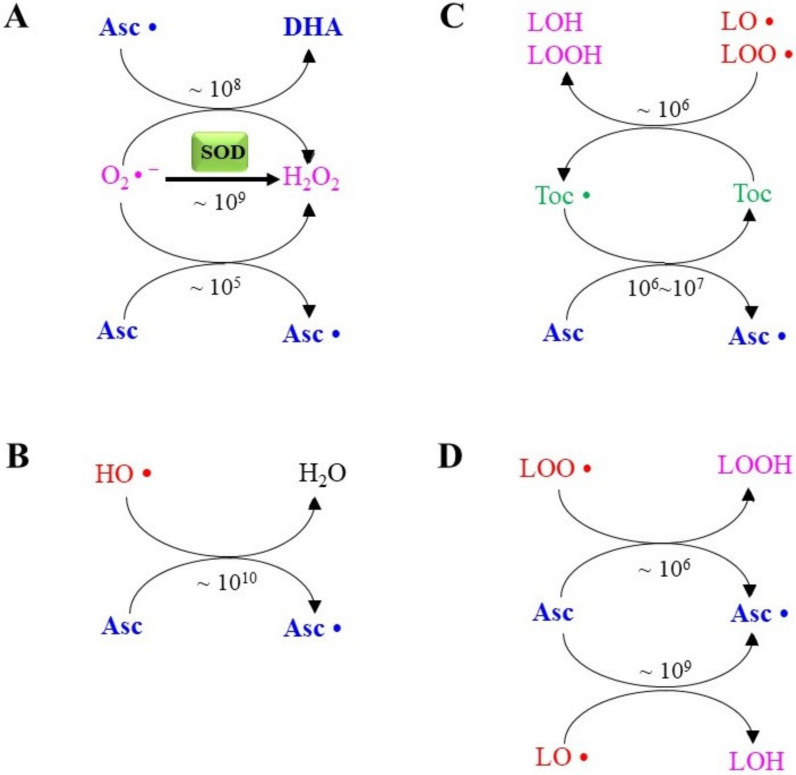
Reaction of Asc with radical species. Asc is involved in the elimination of radical species, as typified by O_2_•^−^ (**A**), HO• (**B**), Toc• produced by the reaction with LO (O)• (**C**), and LO (O)•directly (**D**). The catalytic dismutation by SOD is shown in (**A**) as a reference. Asc• exhibits stronger radical-scavenging ability than Asc, but, because of its abundance, Asc-mediated reactions appear to be feasible in vivo. Numerical values of rate constants (M^−1^s^−1^) for reactions are shown.

**Figure 3 molecules-27-06187-f003:**
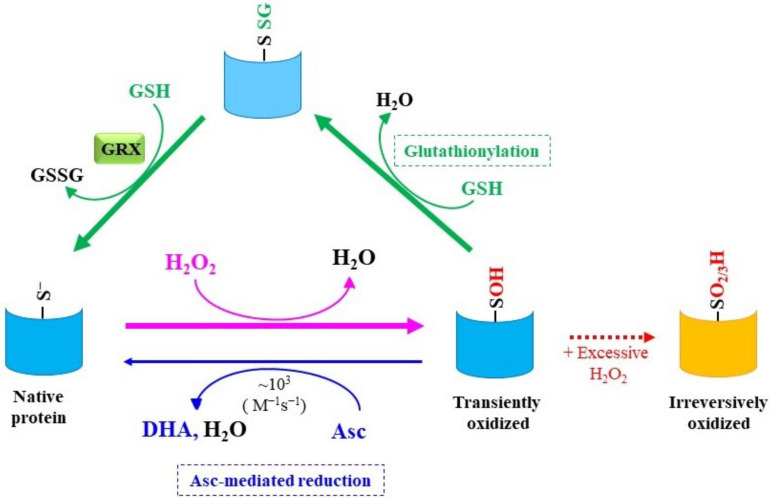
Reductive recycling of Cys-SOH in proteins. The oxidation and reductive rescue of proteins without the involvement of peroxidase is schematized. Reactive Cys in certain proteins can be ionized to Cys-S^−^, which is preferentially oxidized to Cys-SOH by the action of H_2_O_2_. Cys-SOH tends to react with GSH to form a mixed disulfide (Cys-SSG) because of abundant presence of GSH within cells. GRX can reduce it back to Cys-SH by means of another GSH and result in GSSG. Asc may reduce Cys-SOH back to Cys-SH, which proceeds in a non-enzymatic manner. The presence of excessive H_2_O_2_ may further oxidize Cys-SOH to Cys sulfinic acid (Cys-SO_2_H) and Cys sulfonic acid (Cys-SO_3_H), which are the permanently oxidized forms. Overall reactions constituting the oxidation of Cys-S^−^ by H_2_O_2_ and subsequent reduction by Asc consumes H_2_O_2_, resulting in the formation of water and DHA.

**Figure 4 molecules-27-06187-f004:**
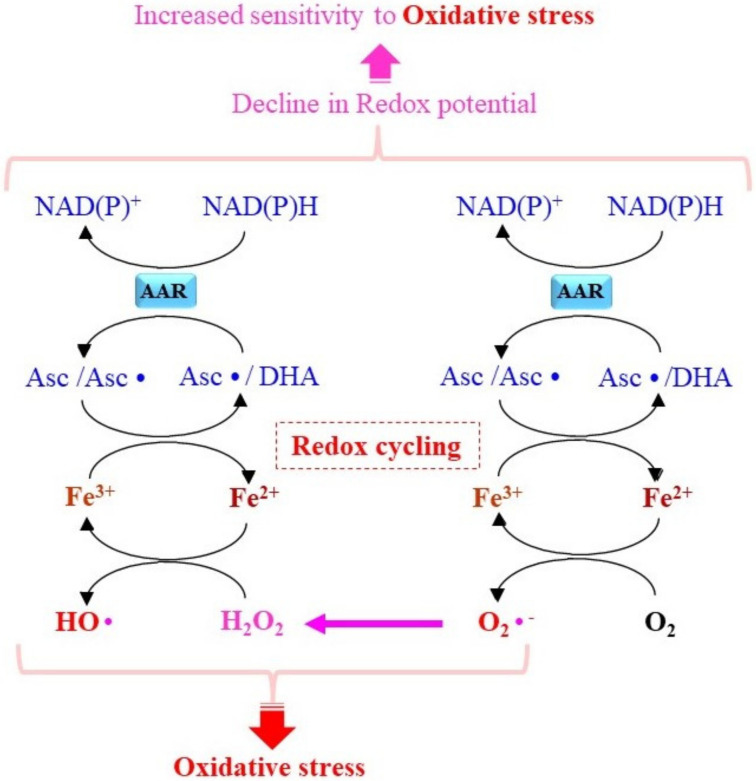
Pro-oxidant effects of Asc in the presence of free iron. Transition metal ions may be involved in ROS production, and, because of its abundance, iron is the most potent metal ion. Fe^3+^ is reduced to Fe^2+^ by accepting an electron from Asc. The resulting Asc• or DHA can be reduced back to Asc by AAR in an NADPH-dependent manner. Because NADPH is also an electron donor for a variety of enzymes, many of which are involved in redox reactions, the consumption of NADPH decreases the redox potential of cells. Fe^2+^ can convert O_2_ to O_2_•^−^ which then undergoes dismutation to H_2_O_2_ and molecular oxygen via either spontaneous or SOD-catalyzed reaction. Reaction of the resulting H_2_O_2_ with Fe^2+^ produces HO•, the most reactive oxidant. This cyclic reaction proceeds as long as Asc and free iron are present, leading to oxidative damage.

**Figure 5 molecules-27-06187-f005:**
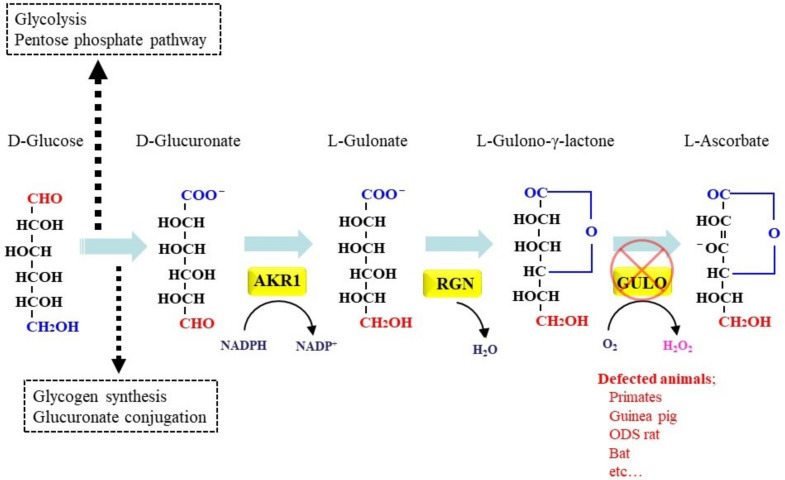
Metabolic pathways for the synthesis of Asc from Glc along with related pathways. Structures of Glc, Asc, and intermediate compounds in the last three processes of the Asc-synthesizing pathway are shown. Dotted lines indicate alternate carbon flows from Glc that may proceed coordinately with Asc synthesis. Three enzymes, AKR1, RGN, and GULO, that are involved in the last three steps of the Asc-synthesizing pathway, are depicted. Mutations in the GULO gene during evolution have rendered some animals incapable of synthesizing Asc.

## Data Availability

Not applicable.

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
