# Peer review of "Ascorbate Is a Primary Antioxidant in Mammals"

_molecules, 2022, doi:10.3390/molecules27196187_

Round 1

Reviewer 1 Report

The content of the manuscript is suitable and suitable for publication.

There are some grammatical errors. Please double-check it.

Author Response

Thank you for your favorable comments. We have reexamined again and amended the manuscript.

Reviewer 2 Report

This review focuses on transgenic mouse models to decipher the physiological role of Asc. Those models reveal unexpected results that could question the physiological role of asc. 

The review is well written and really informative. The future perspective should more emphasize the unexpected results that should prompt us to increase researches on the physiological role of Asc. Moreover there is a typo error in this paragraph.

Author Response

Thank you for your favorable comments. We have revised Future Perspective by adding following statements that emphasize results and critical issues on Asc and will promote researches on the physiological role of Asc .

“By employing these mice, we could obtain new knowledge that were not obtained from studies on in vitro chemical reactions or cultured cells. For example, regarding the in vivo action of Asc as a cofactor for enzymatic reactions, the contribution of complementary molecules that act as electron donors has emerged. On the other hand, the usefulness of Asc as an antioxidant has been found to be more important than previously assumed. However, the use of genetically modified mice faces several challenges to overcome. Mouse metabolism is significantly different from humans, hence, caution should be taken in extrapolating the results obtained from mice to understanding human physiology or pathology. There are also limitations to the surgical procedures that can be performed when creating disease models in mice.”

We also have read through and amended the manuscript several times.

Reviewer 3 Report

In the manuscript entitled "Ascorbate is a primary antioxidant in mammal" the author described the several functions of ascorbate, its biochemistry, and its therapeutic uses. In general, the manuscript is well organized and easy to follow. Proofreading might help improve the quality of the paper, I observed a few typos and grammar issues.

I just have one specific request for the authors to review. In the third paragraph of the introduction, the authors wrote "...we attempt to verify roles of Asc..." the use of the word verify implies that they would be doing experiments, this is a review and they are not presenting evidence that they produced. I would recommend revising the statement.

Author Response

Thank you very much for kindly advice to revise the statement. We have rephrased the "...we attempt to verify roles of Asc..." to “...we attempt to overview roles of Asc...” according to the reviewer’s advice.

We also have read through and amended the manuscript several times.